# IVRIT.AI
# A Comprehensive Dataset of Hebrew Speech for AI Research and Development

## Abstract

We introduce *ivrit.ai*, a comprehensive Hebrew speech dataset, addressing the distinct lack of extensive, high-quality resources for advancing Automated Speech Recognition (ASR) technology in Hebrew. With over 10,000 speech hours and over a thousand diverse speakers, *ivrit.ai* offers a substantial compilation of Hebrew speech across various contexts. It is delivered in three forms to cater to varying research and development needs: raw unprocessed audio; data post-Voice Activity Detection, and partially transcribed data. The dataset stands out for its legal accessibility, permitting use at no cost, thereby serving as a crucial resource for researchers, developers, and commercial entities. *ivrit.ai* opens up numerous applications, offering vast potential to enhance AI capabilities in Hebrew. Future efforts aim to expand *ivrit.ai* further, thereby advancing Hebrew's standing in AI research and technology.

## 1 Introduction

Automated Speech Recognition (ASR; also referred to as speech-to-text) technology holds vast potential for enhancing various processes involving human speech. Nevertheless, its effectiveness is not uniformly distributed across languages. While some languages significantly benefit from ASR tools, others, such as Hebrew, find these technologies to be underwhelming. This often results in the continued use of human scribes when transcriptions are not available.

The initial phase of creating precise and efficient ASR tools requires an extensive corpus of high-quality speech data. To our knowledge, such a dataset for Hebrew has not been publicly available until now. We introduce *ivrit.ai* ("Ivrit" is how Hebrew speakers pronounce the name of their language), a comprehensive Hebrew speech dataset designed for AI research and development. *ivrit.ai* is designed to empower advanced AI technologies to function smoothly in Hebrew, enabling AI to read, write, listen to, and articulate the language with fluency. We view this as a strategic initiative to provide Hebrew speakers and their communities with access to superior AI technology in a practical manner under a favorable license, thus enabling interested commercial parties to use this data at no cost. *ivrit.ai* contains approximately 10,000 hours of Hebrew speech, collected from a diverse range of online platforms including podcasts and other audio content. Given the wide variety of speech types and topics within *ivrit.ai*, it serves as a valuable resource for advancing AI in Hebrew.

Furthermore, the development of *ivrit.ai* is not only a step forward for Hebrew language technology but also a potential blueprint for enhancing ASR tools in other languages with fewer speakers or fewer transcribed audio sources. By setting new milestones in data collection, licensing, and algorithmic design, this project can serve as a foundational model for minority or less-represented languages seeking to leverage ASR technology. The open-access nature and favorable licensing of *ivrit.ai* offer a viable framework for public-private partnerships, paving the way for resource-strapped linguistic communities to accelerate their own AI initiatives. Furthermore, the codebase and methods employed in *ivrit.ai* can be adapted to develop robust ASR tools for other languages, thus promoting linguistic diversity in AI.

## 1.1 Automated Speech Recognition

Oral communication is a fundamental aspect of human interaction, with nuances like pitch, pace, and non-verbal cues making it rich and complex. Humans handle these intricacies naturally, but ASR technologies face significant challenges in replicating this ease. These challenges range from physical complexities like accents and background noise to cognitive aspects such as understanding contextual meanings and appropriate punctuation. While prior research has employed algorithmic approaches to tackle these issues, they often suffer from limited generalization across different languages, speakers, and environments, impacting their real-world effectiveness.

## 1.2 ASR in the age of Large Language Models

Large language models (LLMs; e.g., ChatGPT by OpenAI Roumeliotis & Tselikas (2023), Bard by Google Manyika (2023)) are revolutionizing the tools we use and also hold promise for the ASR task. Relying on transformer-based architecture Zhao et al. (2023) and vast datasets, LLMs are currently recognized as state-of-the-art (SOTA) for typical natural language processing (NLP) tasks (e.g., Hendy et al. (2023), Alarcon et al. (2021), Luo et al. (2022)).

The remarkable success of LLMs in text-based tasks has sparked interest in the field of speech processing to develop solutions for the ASR deficits mentioned earlier, based on similar principles: transformer-based architecture Latif et al. (2023) and large corpora. Currently, ASR models based on these principles (e.g., Whisper Radford et al. (2023), SpeechT5 Ao et al. (2021)) are considered state-of-the-art (SOTA) for most ASR tasks.

To fully realize the significant benefits of ASR and its wide range of applications, we essentially need large, high-quality datasets. These datasets should include a wide vocabulary, diverse speakers, and a variety of topics. Since languages differ in multiple aspects (e.g., phonetics, syntax, and semantic structure), one may utilize datasets tailored to each specific language the ASR should support. At present, only a few languages have access to datasets of sufficient size and quality. Many languages suffer from a lack of resources, which prevents their speakers from maximizing the potential of existing ASR technology. In these languages, a lack of resources poses a major barrier to the effective adoption and utilization of ASR technology. In light of these challenges, it's crucial to support languages in the digital world to foster linguistic diversity and inclusivity. Making more speech and transcribed speech data available for use could significantly benefit less-prevalent languages. Specifically, in this project, we aim to provide such support for Hebrew speakers.

## 1.3 Processing Hebrew Speech

The processing of Hebrew speech is challenged by the lack of available data. Some new monolingual datasets do exist in Hebrew (Sharoni et al. (2023)) and there are some multilingual datasets that include Hebrew Black (2019). However, their scope and quality do not meet the requirements needed for training and optimizing robust ASR models. As a result, Hebrew speakers cannot fully benefit from advancements in ASR technology. It is crucial to collect and develop comprehensive datasets in Hebrew.

## 1.4 The Present Dataset

The *ivrit.ai* dataset we present here has the potential to contribute substantially to various speech and text tasks in Hebrew. This dataset includes over 10,000 hours of Hebrew speech, collected from multiple online sources. The dataset includes a wide range of speakers, varying by gender, origin, and education level, as well as a variety of speech styles (e.g., official lectures, informal conversations), and topics (e.g., sports podcasts, Talmud lessons). Approximately 8.4 million utterances and about 90 million words are included in the dataset. The audio data and transcripts are available for research and development in AI modeling, encompassing both non-commercial and commercial uses.

## 2 RELATED WORKS

Significant advances in ASR and NLP have been made in recent years, as a result of numerous datasets and research studies. However, the Hebrew language has received very little attention in this domain.

### 2.1 GENERAL SPEECH DATASETS AND RESEARCH

Monolingual speech datasets have been developed for specific domains and communication situations, including the ATIS corpus for air travel information requests Hemphill et al. (1990), as well as other corpora containing recordings of meetings Garofolo et al. (2004b), telephone conversations Canavan et al. (1997), and broadcast media Garofolo et al. (2004a). The original TIMIT collection Garofolo (1993) and many of the broadcast news corpora provide relatively clean audio scenarios in which a single speaker reads from a prepared text, such as TREC Garofolo et al. (2000) and CLEF Federico & Jones (2004). This characteristic restricts the usefulness of the data in more dynamic environments and in more spontaneous situations. There are also collections of more naturally occurring conversational material, such as the CALLHOME corpus Canavan et al. (1997), the Santa Barbara Corpus of Spoken American English Du Bois et al. (2000), and the TED talks corpus Hasebe (2015), as well as podcast corpora, for instance, Spotify Clifton et al. (2020) and MSP Lotfian & Busso (2017) Martinez-Lucas et al. (2020). These collections capture unscripted and spontaneously organized discourse in a conversational setting, including turns, interviews, stretches of monologue, and argumentation. Due to this aspect, this data is more useful for dealing with natural language patterns and real-world dialogue dynamics. None of these datasets include Hebrew speech.

### 2.2 MULTILINGUAL SPEECH DATASETS AND RESEARCH

Current multilingual speech datasets with transcriptions, such as those drawn from conversational telephone speech (IARPA Babel Program Cui et al. (2013)), political speech Wang et al. (2021), and audiobooks Panayotov et al. (2015) Pratap et al. (2020), cover a variety of domains and span approximately 100 languages. However, the representation of Hebrew in these datasets is relatively low. This limitation also extends to multilingual datasets without transcriptions, such as VoxLingua107 Valk & Alumäe (2021) and VoxPopuli Wang et al. (2021), which span multiple languages and contain large amounts of unlabeled data. Even in datasets like the one utilizing read versions of the New Testament (MSU Black (2019) and recently MMS Pratap et al. (2023)), which cover many languages and provide high-quality alignments, the Hebrew content is not extensive. Unfortunately, the authors don't detail the Hebrew content, but generally suggest around 25 hours per language. The data from these datasets is used to train self-supervised models, build speech recognition systems, and develop language identification models Pratap et al. (2023) Wang et al. (2021). Despite the availability of numerous monolingual and multilingual speech datasets, each with their unique characteristics, limitations, and areas of focus, the representation of Hebrew remains limited. The field continues to evolve with ongoing efforts to develop more comprehensive and diverse datasets that can support a wider range of ASR and NLP research. Thus, there is a clear need for more extensive Hebrew content in these resources.

### 2.3 PRIOR HEBREW SPEECH DATASETS

An array of datasets spanning academic to non-academic ventures provide resources for Hebrew ASR research. Part of the datasets are monolingual, containing Hebrew exclusively, and part of them are multilingual, encompassing Hebrew along with other languages. Table 1 offers an overview of these available Hebrew speech datasets, both mono and multilingual, outlining their distinct characteristics and inherent limitations. In comparison to existing datasets, *ivrit.ai* sets itself apart by offering a comprehensive corpus with over 10,000 hours of audio, featuring more than 1,000 speakers and covering a wide array of topics. Additionally, our augmented CC BY 4.0 license promotes both academic research and commercial usage.

Table 1: Comparison of various speech datasets

| Corpus | Hours | Speakers | Trans. | Type | Topics | License |
|---|---|---|---|---|---|---|
| SASPEECH Sharoni et al. (2023) | 30 | 1 | 4M/26A | Mixed | Economy, Politics | Non-commercial |
| HUJI Corpus Marmorstein & Matalon (2022) | 3.8 | 60 | M | Conversations | General Lifestyle | CC BY 4.0 |
| CoSIH Izre'el et al. (2001) | 12.3 | ±140 | M | Conversations | General Lifestyle | Non-commercial |
| MaTaCOp Azogui et al. (2016) | 5.3 | 16 | M | Conversations | Map Task framework Anderson et al. (1991) | Non-commercial |
| MLS Pratap et al. (2020) | 126 | 13 | M | Reading | Classic Books | CC BY 4.0 |
| CMU Black (2019) | ±25 | - | M | Reading | New Testament | - |
| MMS Pratap et al. (2023) | ±25 | - | M | Reading | New Testament | - |
| Whisper Radford et al. (2023) | 688 | - | - | - | - | Not available |
| ivrit.ai | +10,000 | +1000 | A | Mixed | Wide range (economy, politics, science, bible, philosophy, technology, history, etc.) | Augmented CC BY 4.0 (see Availability section) |

The columns show the name of the corpus, total hours of audio available, the number of speakers included, whether the dataset is transcribed or not (M for manual transcription, and A for automatic transcription, the type of speech included (reading, conversations, mixed), the topics covered in the dataset, and the terms of the license for using the dataset. Dash (-) indicates that the data is not available for the corresponding field.

## 2.4 NATURAL LANGUAGE PROCESSING AND SPEECH RECOGNITION RESEARCH IN HEBREW

Substantial progress has been made in the field of Hebrew NLP, with models like Hebert Chriqui & Yahav (2022), AlephBERT Seker et al. (2021), and AlephBERTGimmel Guetta et al. (2022) setting benchmarks in various textual tasks. These advancements illustrate the potential for an intricate understanding of written Hebrew texts.

The current landscape of Hebrew Automatic Speech Recognition (ASR) is largely supported by commercial ASR services like Google Cloud, Microsoft (e.g., via Word), Samsung, IBM Watson, and WIT.ai. These services offer engines with accessible APIs that support Hebrew Silber-Varod et al. (2021). Additionally, openly accessible models like OpenAI's Whisper Radford et al. (2023) and Meta AI's multilingual model Pratap et al. (2023) are available as open sources. This availability allows developers to create their own models for specific tasks. Expanding on these resources, we have contributed to Hebrew ASR development by introducing the *ivrit.ai* dataset. This open-access dataset, free for all interested parties, equips researchers and developers with the means to further improve their Hebrew ASR models.

## 2.5 UNLABELED SPEECH DATASETS AND RESEARCH

In the field of ASR, significant progress has been achieved through the application and evolution of unsupervised speech pre-training techniques Wu et al. (2020), semi-supervised learning (self-training) Liu et al. (2022), and the combination of these techniques Xu et al. (2021). However, the

research community focuses primarily on ASR tasks utilizing English as the main input. Through their proficient use of a copious amount of unlabeled English speech data, these methods have improved English-centric ASR applications. A recently published unlabeled speech corpus, covering 23 languages, demonstrated the potential of these techniques for more languages Wang et al. (2021). However, Hebrew was not included in this corpus. These results underscore the urgent need for large datasets of Hebrew speech, even if they are not labeled. Significant progress can be expected in Hebrew ASR if these datasets are developed and utilized effectively.

## 3 DATASET CREATION

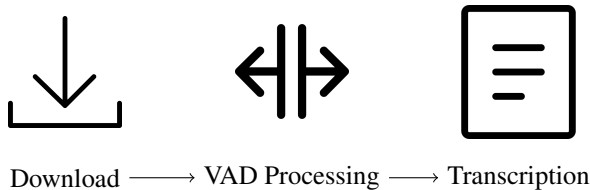

Download $\longrightarrow$ VAD Processing $\longrightarrow$ Transcription

Figure 1: Illustration of the Data Pipeline: (1) Downloading the data in accordance with data creator agreements, (2) Processing Voice Activity Detection (VAD) to segment audio based on silences, and (3) Transcribing the segmented audio, which currently relies solely on machine-based process but is planned to incorporate human transcription.

The data and code are openly available at Hugging Face (`https://huggingface.co/ivrit-ai`) and the GitHub repository (`https://github.com/yairl/ivrit.ai`). Throughout this section, we will describe the process of creating *ivrit.ai*. This resource, with its variety of speakers and audio qualities, offers a comprehensive representation of the Hebrew language in multiple contexts. Detailed information on how it was collected and preprocessed will be provided. Figure 1 shows a schematic diagram of the dataset creation pipeline.

### 3.1 DATA ACQUISITION

We gathered audio clips from a variety of sources, encompassing both individual and institutional content creators.

ivrit.ai's license is specifically designed to enable commercial use of this corpus for training AI models, such as speech-to-text or LLM models, while preserving the intellectual property rights of the content owner. Every item in the corpus has been provided by its content owner who signed permission to use this item under ivrit.ai's license, thereby permitting the use of their work in this manner.

The advantages of such an agreement are twofold. First, it ensures fairness towards the content creators by explicitly stating in the agreement that their work can be used. Second, it provides certainty for research and development entities by confirming that the data has been collected with permission and is distributed within a suitable legal framework.

### 3.2 DATA PROCESSING

The *ivrit.ai* data we've collected are being released in three distinct datasets, catering to various research needs:

- Raw data: This is the original, unprocessed collection of audio clips.
- Data post-VAD: For this dataset, we have run voice activity detection (VAD) on the raw audio Team (2021). This operation segregates short units, ranging from a few seconds to a minute, pinpointing parts where speakers were actively involved. Figure 2 provides insights into the length distribution of the audio pieces post-VAD.
- Partially transcribed data: This dataset provides a portion of the audio data along with their corresponding transcriptions.

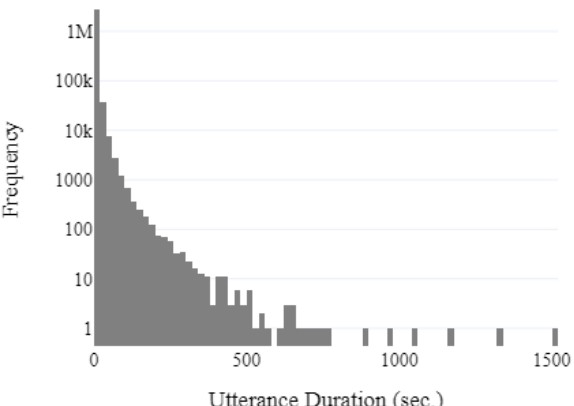

Figure 2: Distribution of post-VAD audio clips in *ivrit.ai* corpus. The x-axis represents the segment length in seconds, while the y-axis indicates the frequency of each duration range

### 3.3 Transcribed Speech

The *ivrit.ai* dataset has been transcribed using the Whisper ASR tool (Radford et al. (2023), using the *whisper-small* model). The transcription process is applied to numerous short audio segments, resulting in 2.8 million transcribed utterances and about 90 million words.

## 4 Dataset Description

As the data was collected from multiple contributors and represents a wide range of recordings (narrative podcast, conversation, lesson), our possibility to provide precise information regarding the speakers is limited. However, we can provide some general information. The *ivrit.ai* dataset comprises over 10,000 hours of speech from a thousand diverse speakers. The dataset encompasses a wide range of audio file lengths, and Figure 3 provides insights into the distribution of episode lengths within the dataset.

Speakers' ages ranged between 20 to 90 years. While some speakers are native Hebrew speakers, for others, Hebrew is the second language (with English, Russian, Arabic, and other languages being their native languages), adding multiple accents to the spoken languages.

Languages other than Hebrew, predominantly English, appear within the corpus at three levels of magnitude. First, as single words borrowed for specific uses, technical or otherwise; second, as slightly longer phrases that have gained popularity in Hebrew usage (for instance, the phrase "having said that" has been adopted into Hebrew conversations as is); and third, there are entire episodes that are conducted solely in English.

## 5 Availability

The dataset is publicly accessible on the *ivrit.ai* website and is distributed under an *ivrit.ai* license, an augmented CC-BY 4.0 license tailored to allow AI model training for commercial use. Detailed licensing terms can be found in Appendix A.

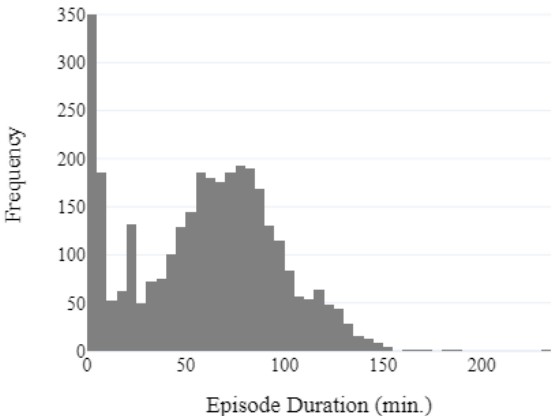

Figure 3: Histogram depicting the distribution of episode durations in *ivrit.ai* corpus. The x-axis represents the episode duration in minutes, while the y-axis indicates the frequency of each duration range

## 6    DISCUSSION

We present *ivrit.ai*, a comprehensive collection of over 10,000 hours of high-quality Hebrew speech, curated to advance AI research in the Hebrew language. This novel dataset consists of a wide range of speech types and topics, ready for use in various applications such as emergency response systems, accessibility tools for the disabled, medical transcription services, and digital voice assistants in the service industry, among others.

The *ivrit.ai* dataset stands out among other Hebrew datasets in its size, diversity, and coverage of different speech styles and domains. Furthermore, the *ivrit.ai* dataset offers more legal accessibility than many other datasets, not just in Hebrew but also in other languages, as it is available for industrial use, making it a valuable resource for researchers and developers. Researchers can leverage this dataset to train and evaluate their models, while also utilizing it as a benchmark for performance comparison. Importantly, our methodology, milestones, and legal framework developed for *ivrit.ai* could serve as a blueprint for addressing similar challenges in other languages that are currently underserved.

Among the limitations of the project, the dataset may be biased in aspects such as gender or age imbalances, which may affect the performance of AI models trained on the dataset. Additionally, the dataset's diversity, considered above as a strength, introduces variability in recording means, speaker characteristics, and background noises. Moreover, deficits in data collection and transcription could impact the dataset's quality or usability.

## 7    CONCLUSION AND FUTURE WORK

ASR technology holds vast potential for enhancing various human processes. Although generating high-quality, efficient ASR tools is well known, model quality depends on the dataset size. Despite the benefits that can be obtained from ASR tools for some languages, others, such as Hebrew, are underwhelmed by the technology.

We introduced the *ivrit.ai* dataset, a comprehensive collection of Hebrew speech, designed to advance AI research in Hebrew. In our view, the availability of such a diverse and extensive dataset is a significant step forward in the field of Hebrew ASR and NLP research. This dataset has the potential to improve multiple ASR-based systems' accuracy and performance.

Dataset acquisition tends to be effort-intensive, and fraught with legal difficulties due to copyright requirements that often conflict with standard licenses. *ivrit.ai* aims to create the world's largest freely-available audio dataset in Hebrew, fully transcribed, and fully available for the specific purpose of training ASR and AI models.

Looking forward, we plan to further expand the *ivrit.ai* dataset, increase the corpus by another order of magnitude and promote applied developments based on the dataset, particularly in specific domains. Additionally, we intend to publish models for common speech tasks such as Speech-to-Text (STT), Text-to-Speech (TTS), speaker diarization, etc. Our aim is to place Hebrew at the forefront of AI research and technology by making the current dataset and future models widely accessible.

### ACKNOWLEDGMENTS

We would like to express our deepest gratitude to all the content creators who generously allowed us to use their data for this project. Their contributions have been invaluable in advancing AI research in Hebrew. The full list of data contributors is updated and available on the *ivrit.ai* website.

We also extend our heartfelt thanks to Adv. Eli Greenbaum from Yigal Arnon & Co., who generously provided his legal expertise pro bono to draft the license for this open data project. His contribution has been instrumental in ensuring the accessibility and wide distribution of the *ivrit.ai* dataset.

Your collective support and contributions have been instrumental in the success of this project, and we look forward to seeing the advancements in AI research that the *ivrit.ai* dataset will facilitate.

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

## A    LICENSE APPENDIX

