# OpenReview forum: "ivrit.ai: A Comprehensive Dataset of Hebrew Speech for AI Research and Development"
_ICLR.cc/2024/Conference — Submitted to ICLR 2024_

### Official Review · Reviewer_toHs · 2023-10-25

**Soundness:** 3 good
**Presentation:** 3 good
**Contribution:** 2 fair
**Rating:** 3
**Confidence:** 3

**Summary:**

This paper introduces a novel ASR dataset for Hebrew.  The dataset consists of 10k hours of audio with VAD and some automatic transcription.

**Strengths:**

The paper's claim that this dataset much larger and much more diverse than any previous Hebrew ASR dataset is to my knowledge correct.  I further agree with claim that the contribution of such a dataset is of profound importance to research in Hebrew ASR and to the development of strong speech products for Hebrew.  Specific strengths include:

- The dataset is indeed under a permissive license.
- The dataset includes speakers of many ages and include speakers with other first languages representing the diversity of Hebrew speakers as a whole.
- The settings from which the data is reported to have been gathered reflect realistic applications of Hebrew ASR.

**Weaknesses:**

The importance of the contributed dataset is clear, and I offer no challenge to it.  However, I believe that the write-up is missing a few things that users will need and will expect to find in a paper that introduces a dataset.

(1) Details on how the dataset is split across speakers and domains.  To whatever degree is possible, information on how much of the data came from speakers in different age groups, speakers with different first languages, and settings (podcasts, lectures, talmud lessons, ect.)

(2) Details on the automatic transcription.  How much of the 10k hours are transcribed?  How good is the transcription? Even a WER on a very small (maybe 30mins?), manually transcribed test split would be helpful as evidence that the transcriptions are strong enough to be usable.

(3) Baseline models.  How does a conventional RNN-T or LAS model perform when trained on the supervised subset?  How about when it is then finetuned using the untranscribed subset, or rescored with a language model?

For an example of a classic and very useful paper of this kind, I direct the authors to the original Librispeech write-up: https://www.danielpovey.com/files/2015_icassp_librispeech.pdf

**Questions:**

Please see the three points above.  I believe that after even partially addressing those missing pieces this write-up will be a substantial contribution.

---

### Official Review · Reviewer_7nK7 · 2023-10-30

**Soundness:** 2 fair
**Presentation:** 3 good
**Contribution:** 1 poor
**Rating:** 1
**Confidence:** 5

**Summary:**

The paper presents a new, open-source dataset consisting of Hebrew audio files, partially transcribed by an automatic speech recognition system (Whisper). The paper furthermore provides an overview of other Hebrews speech resources (most of the paper), and some details regarding data collection.

**Strengths:**

The authors provide a new resource to the speech research community - there has been a lack of publicly available corpora for Hebrew and the ivrit.ai corpus fills a gap.

**Weaknesses:**

The paper only describes a data collection effort; it does not present a research hypothesis or experimental study. As such it is not well suited to ICLR - a conference like Interspeech or LREC with a track for language resources would be more appropriate.
Even for a speech or language resources conference, it would be desirable to have more details on the actual corpus, such as quantitative information about language/accent distribution.
Lastly, the corpus would be much more valuable if some amount of human annotation were included. Automatic transcription contains errors, and especially systems like Whisper are known to hallucinate. These errors will be propagated if users of this corpus use the transcripts for further model training. Automatic transcription is easy to generate, but manual transcription is difficult and costly -- it would have been a great service to the community to include even a few hours of manual transcripts.

**Questions:**

The paper states that "our possibility to provide precise information regarding the speakers is limited" and therefore, no distribution across speaker groups is provided. However, you do seem to know that speakers range between 20 and 90 years. If this information is available, why can't a distribution over age ranges be provided? It would be good to see the distribution over ages, accents, and other languages, even if based on automatic classifiers.

---

### Official Review · Reviewer_PchH · 2023-10-30

**Soundness:** 2 fair
**Presentation:** 2 fair
**Contribution:** 2 fair
**Rating:** 3
**Confidence:** 4

**Summary:**

The paper introduces ivrit.ai, which is a comprehensive Hebrew speech dataset designed to address the lack of extensive and high-quality resources for Automated Speech Recognition (ASR) technology in Hebrew. The dataset consists of over 10,000 speech hours from a diverse set of speakers, covering various contexts. It is provided in three different forms: raw unprocessed audio, data post-Voice Activity Detection, and partially transcribed data. What sets ivrit.ai apart is its legal accessibility, allowing free usage, making it a valuable resource for researchers, developers, and commercial entities. The dataset has the potential to enhance AI capabilities in Hebrew.

**Strengths:**

Language-specific datasets play a crucial role in advancing ASR technology for specific languages. Hebrew, as a complex and unique language, presents its own set of challenges and nuances in speech recognition. By addressing the distinct lack of extensive and high-quality resources for Hebrew ASR, the paper fills an important gap in the research landscape.

**Weaknesses:**

The primary concern with the paper is its narrow focus on Hebrew ASR dataset. Given the diverse range of topics covered at ICLR, which includes machine learning, representation learning, and various other areas, a paper solely dedicated to a specific language may struggle to attract a broad audience. The conference typically prioritizes research with broader applicability and impact across multiple domains. Although the paper addresses the need for extensive and high-quality resources for Hebrew ASR, the impact beyond the Hebrew language itself seems limited. While Hebrew is a unique language with its own complexities, the specific challenges faced in Hebrew ASR may not resonate with researchers working on other languages or broader speech recognition topics. This lack of broader impact may further diminish the paper's appeal to the ICLR audience. I suggest the authors submit the paper to specific workshop or speech conference.

 Another concern is the absence of baseline numbers with well-established open-source frameworks such as ESPnet, k2, and fairseq. Others could easily build their system and have a relatively fair comparision if these numbers are provided.

**Questions:**

Why not use Whisper large to generate transtriptions?

---

### Official Review · Reviewer_EH9S · 2023-11-01

**Soundness:** 1 poor
**Presentation:** 2 fair
**Contribution:** 1 poor
**Rating:** 3
**Confidence:** 5

**Summary:**

This work introduces ivrit.ai, which is a large Hebrew speech corpus consisting of more than 10k hours. The authors apply several preprocessing approaches to clean and organize the dataset.

**Strengths:**

The corpus has more than 10k hours of Hebrew speech, which is likely to be very valuable for future research on Hebrew ASR.

The corpus also has CC-license, which makes it available to many use cases.

**Weaknesses:**

Although the corpus should be valuable to the community, this work does not implement any baseline models or provide any sufficient qualitative analysis of the collected speech. This makes it very difficult to assess the quality of this corpus and understand the underlying contents.

Establishing a good baseline of the corpus should be conducted, for example, adapting Whisper to this corpus.

**Questions:**

Can authors give more details about how this corpus is collected?

---

### Meta-Review · Area_Chair_q9Wr · 2023-11-29

**Metareview:**

This paper introduces the "ivrit.ai" Hebrew speech dataset which is helpful for the speech community for speech-related research and commercialization.  Despite the importance of the topic, the paper in its current form needs significant improvements in order to get accepted.  All reviewers raised concerns such as missing important information on some technical  details on constructing the dataset and decent ASR baselines typically coming along with a release of a dataset.  Since no rebuttal is provided by the authors, these concerns                   are not cleared.

**Justification For Why Not Higher Score:**

The paper needs significant improvements to get accepted given its current form.  No rebuttal is provided.

**Justification For Why Not Lower Score:**

N/A

---

### Decision · Program_Chairs · 2024-01-16

Reject